# A novel method for generating public keys involving matrix operations

Xin Sun[1], Jiajia Han[1], Bang Lv[1], Changhua Sun[1], Cheng Zeng ᴵᴰ[2]*

1 State Grid Zhejiang Electric Power Co., Ltd. Research Institute, Hangzhou, Zhejiang, P. R. China, 2 School of Mathematical Sciences, Beihang University, Beijing, P. R. China

* zpr112358@buaa.edu.cn

**Data Availability Statement:** All relevant data are within the manuscript.

**Funding:** Xin Sun, Jiajia Han, Bang Lv, Changhua Sun are supported by State Grid Zhejiang Electric Power Co., Ltd. Research Institute (No. B311DS23000M)(URL:https://www.zj.sgcc.com.

## Abstract

With the rise of the Internet of things, the limitations of traditional PKI certificate authentication technology have progressively emerged. Hence, identity-based public key algorithms have been proposed. In this paper, we propose a new approach to generate an identity key, named identity public key (IPK). IPK identity key generation protocol is based on SM2 elliptic curve cryptography and random matrix theory. It builds upon the concept of combined public key (CPK) and resolves the linear collusion issue of CPK as well as the authenticity verification problem of the declared public key of the simplified TF-CPK by enhancing the identity mapping approach. Another main aim of this paper is to prove the security of the IPK identity key generation protocol. Roughly speaking, we verify the security of each part of the private key and the security of the composite private key. We also increase the function and performance comparison with other schemes, such as IBC(SM9) and TF-CPK. Our scheme has the lowest computation cost, which demonstrates its rationality.

## 1 Introduction

With the swift advancement of computer and network technology, the demand for secure communication [1] is increasingly extensive, and the constraint of the symmetric cryptosystem in key distribution and key management is no longer compatible with technological development. In 1976, Diffie and Hellman [2] proposed the public key cryptosystem, which solved the problem of key distribution. Nevertheless, the issue of public key distribution has persistently posed a formidable challenge, with its authenticity predicament remaining unresolved, consequently facilitating susceptibility to man-in-the-middle attacks during key exchange, refer to [3, 4]. Therefore, in order to solve the binding relationship between the authenticity of public key and the user, a certificate mechanism, public key infrastructure (PKI) [5, 6], is proposed to realize the secure correspondence between the user's identity and the user's key, which solves the difficult problem of key management. PKI certification authority [7–9] based on certificate mechanism has been widely used and verified in the past three decades and has made great contributions to the development of information security, especially in solving the problem of identity authentication based on personnel for e-government, finance, office and other fields of information development security [10–12].

cn/p1/index.html). The sponsors play a role in research design, analysis and manuscript preparation.

**Competing interests:** The authors have declared that no competing interests exist.

However, with the rise of the Internet of things [13, 14], the subject of security authentication is no longer limited to people, but more involves devices, sensors and other entities, and also extends to the identity authentication and communication security of non-entity objects such as software and microservices [15, 16]. In particular, the Internet of things terminal not only has the requirements of narrowband communication and low-power computing, but also has the characteristics of large number and wide area distribution. In addition, the Internet of things application itself also has business diversity and particularity requirements [17], which makes the security protection of it different from the traditional human-oriented Internet security protection. The limitations of the traditional PKI certificate authentication technology in the Internet of things are gradually emerging. Therefore, the identity-based public key algorithms, such as SM9 [18], PKIot [19], PKI4IoT [20] came into being, and they were applied to the security applications of the Internet of things. A general class of identity-based public key cryptosystems is combined public key (CPK) [21–23]. Very recently, some related schemes, such as efficient privacy-preserving spatial range query over outsourced encrypted data [24], time-controllable keyword search scheme with efficient revocation in mobile e-health cloud [25], and efficient privacy-preserving spatial data query in cloud computing [26], instead of conventional schemes have been proposed.

In this paper, we present a series of identity key generation protocols based on the SM2 public key cryptographic algorithm [27, 28], integrating the identity public key generation protocol. Specifically, our approach, known as identity public key (IPK), consists of three distinct steps:

1. The mapping sequence of the terminal identity was determined through the random generation of public keys for both the identity and terminal, as well as the public key of the key center;

2. The corresponding $h$ ($h \geq 32$) matrix elements which relied on the mapping sequence were selected from the combined public and private key matrices to generate the public and private keys of the identity. Subsequently, the distribution's public and private keys were obtained by combining them with the random public and private keys corresponding to the key center;

3. The public and private keys of the terminal were obtained by computing the random public and private keys generated by the terminal in a covert manner.

The organization of our paper is as follows: Section 2 presents a guide to generating private and public keys; Section 3 delves into the security of IPK; The final section provides the summary.

## 2 Technical design

IPK is a key generation protocol based on the identity of SM2 public key cryptography. It mainly includes matrix generation, private key generation and public key calculation. IPK identity key generation protocol transforms existing public keys into identity based public keys in a combinatorial way to solve the problem of public key distribution and proof. The technical principle is based on the combinatorial characteristics of ECC [29], i.e., the sum of private keys and the sum of the corresponding public keys form a new public and private key pair. Suppose that the sum of the private keys is

$$(r_1 + r_2 + \cdots + r_m) \bmod n = r.$$

**Table 1. Symbols and abbreviations.**

| Alice, Bob | Two users using a public key cryptosystem |
|---|---|
| SM3 | SM3 cryptographic hash algorithm |
| HMAC | Hash-based message authentication code |
| $d_A$ | Private key of Alice |
| $d_B$ | Private key of Bob |
| $\mathbb{F}_q$ | Finite field of order $q$ |
| $E(\mathbb{F}_q)$ | Set of all rational points of an elliptic curve $E$ over $\mathbb{F}_q$, including $O$ at infinity |
| $M, M'$ | Message to be signed |
| $e, e'$ | Output values of cryptographic hash algorithm acting on $M, M'$ |
| $G$ | Base point of an elliptic curve whose order is a prime |
| $H_v$ | Cryptographic clutter algorithm with message digest length of $v$-bits |
| $ID_A$ | Identity of Alice |
| $ID_B$ | Identity of Bob |
| $P_A$ | Public key of Alice |
| $P_B$ | Public key of Bob |

Then the sum of the public keys is given by

$$R_1 + R_2 + \cdots + R_m = R.$$

Clearly, $(r, R)$ forms a new public and private key pair, since

$$R = R_1 + R_2 + \cdots + R_m = [r_1]G + [r_2]G + \cdots + [r_m]G = [r]G.$$

Some of the symbols and abbreviations in this paper are shown in Table 1.

## 2.1 Generation of matrix

The key sensitive parameter of IPK ID key generation protocol is IPK private key matrix, which is important for generating private key and identifying the relationship between private key and IPK ID key. 256-bit random numbers are taken as elements of the private key matrix, and the scale of the private key matrix is $m \times h$, where both $m$ and $h$ are powers of 2 and $m \geq h \geq 32$. The private key is also a 256-bit random number. Although the probability of collision is very small (no more than $2^{-256}$), the collision of private key matrix elements will reduce the security of the system, so the $m \times h$ random numbers of private key matrix are required to be different from each other.

Suppose that we have selected $m \times h$ different 256-bit random numbers $r_{i,j}$, where $i \in \{0, 1, \ldots, m - 1\}$, $j \in \{0, 1 \ldots, h - 1\}$. Each random number $r_{i,j}$ is modeled on the order $n$ of the elliptic curve base point $G$ of SM2, that is,

$$sk_{i,j} = r_{i,j} \bmod n.$$

Set $sk_{i,j}$ be the element in row $i$ and column $j$ of the private key matrix. When the private key matrix is generated, each element $sk_{i,j}$ needs to be compared with each other. Only when they are different, the private key matrix element is valid. Otherwise, the random numbers are generated again to calculate the private key matrix element until all conditions are met.

When the private key matrix element is determined, the corresponding public key matrix element is generated through the calculation of the private key matrix element, that is,

$$PK_{i,j} = [sk_{i,j}]G,$$

where $PK_{i,j}$ is the element in row $i$ and column $j$ of the public key matrix.

The private key matrix is only owned by the key center and is stored in hardware cryptographic devices throughout its entire lifecycle. The public key matrix, on the other hand, is the public key generation protocol for IPK, and the public key parameters are distributed to each terminal in the form of a file and stored separately as an important parameter for the terminal to calculate the public keys of other parties.

## 2.2 Generation of private key

The process in Fig 1 is explained in detail by the process of Alice applying for the private key:

1. When Alice applies for a private key, the SM2 key pair $(r_1, R_1)$ is randomly generated, where $R_1 = [r_1]G$;

2. The private key factor $r_1$ is cached, and $R_1$ and the identity of Alice ($ID_A$) are sent to the key center, which randomly generates a key pair $(r_2, R_2)$, where $R_2 = [r_2]G$;

3. The key center computes Alice's declared public key $R_A = R_1 + R_2$, takes $ID_A$ as the data, then calculates $HMAC_{R_A}(ID_A)$ based on SM3, and gets a 32-byte hash value;

4. The hash value is evenly divided into $h$ groups, and the corresponding value $v_i$ of each group is modular by $m$, that is,

$$I_i = v_i \bmod m;$$

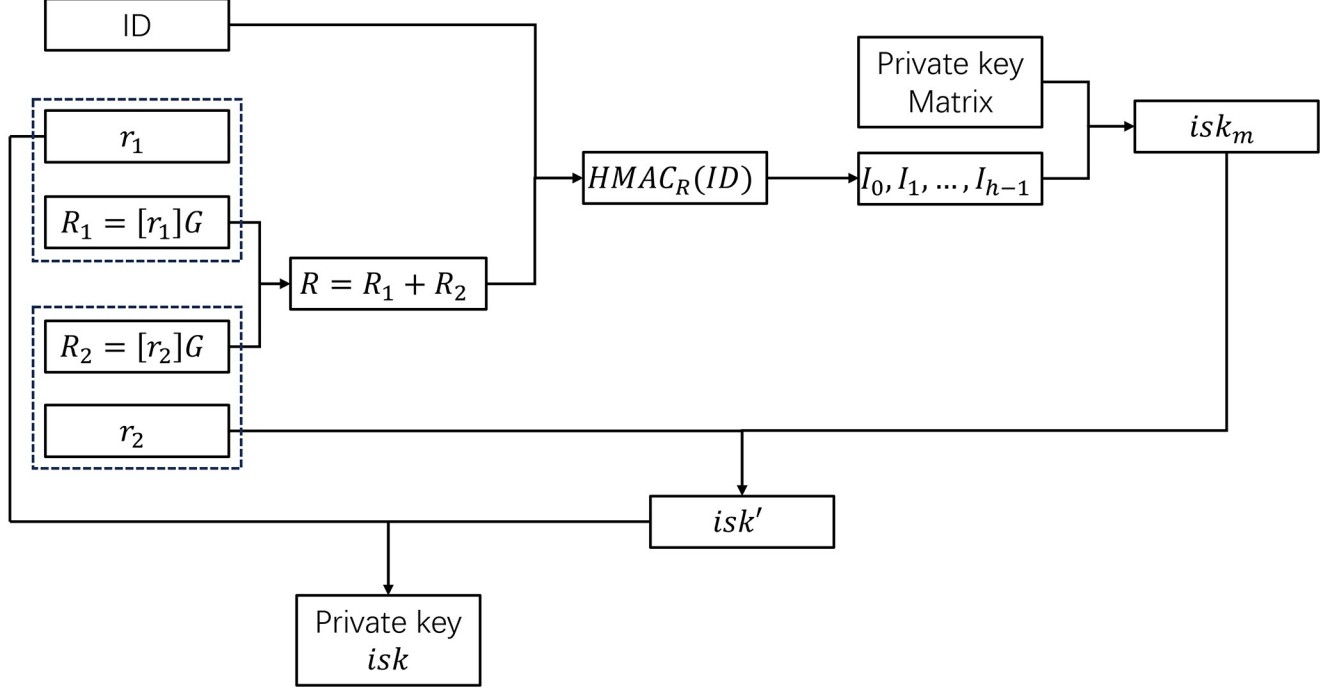

**Fig 1. Process for generating a private key.**

5. Select $h$ private key matrix elements:

$$sk_i = sk_{I_i,i}, i = 0, 1, \ldots, h-1;$$

6. The key center calculates the identity private key

$$isk_m = (sk_0 + sk_1 + \cdots + sk_{h-1}) \bmod n;$$

7. The key center then gives the first-order composite private key

$$isk' = (isk_m + r_2) \bmod n;$$

8. The key center sends $isk'$ and $R_A$ to the applicant Alice through the secure channel;

9. Alice combines the custom private key factor $r_1$ to obtain the second-order composite private key $isk = (isk' + r_1) \bmod n$, and then clears $r_1$.

10. Store the second-order compound private key $isk$ and the declared public key $R_A$.

From the method of generating private keys, the method of generating private keys for IPK is only to calculate the selected coordinates of the matrix based on the identity and the declared public key, and then add the selected matrix elements. This generation method does not affect the security of the SM2 cryptographic algorithm because the generated private key satisfies the randomness principle of the SM2 private key by adding the random numbers $r_1$ submitted by the applicant and $r_2$ generated by the key center.

## 2.3 Computation of public key

Fig 2 describes the generation process of the public key. Alice sends $ID_A$ and $R_A$ to Bob, who can calculate Alice's final public key according to the public key generation algorithm. The specific process is as follows:

1. Bob uses $ID_A$ as the data and $R_A$ as the key to compute the SM3 based HMAC and obtain the 32-byte hash value.

2. The hash value is evenly divided into $h$ groups, and the corresponding value $v_i$ of each group is modular by $m$, that is,

$$I_i = v_i \bmod m;$$

3. Select $h$ public key matrix elements $PK_i = PK_{I_i,i}, i = 0, 1, \ldots h-1;$

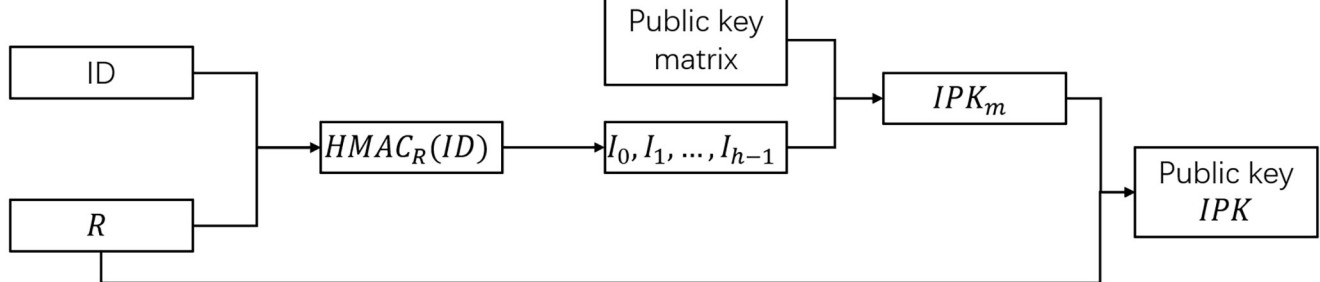

**Fig 2. Process for generating a public key.**

4. Compute identity public key $IPK_m = PK_0 + PK_1 + \cdots + PK_{h-1}$;

5. Calculate the composite public key of Alice $IPK_A = R_A + IKP_m$.

According to the technical principle, the public key generated by the above method is corresponding to the private key, that is, the private key and the public key constitute a key pair. In the generation method, the coordinate selection of the binding generation matrix between the signer Alice's identity $ID_A$ and the declared public key $R_A$ solves the substitution attack problem of the declared public key and realizes the binding relationship between the identity and the declared public key.

## 3 Security analysis

The security of IPK key generation protocol mainly includes four aspects: First, the analysis of whether there is collusion attack risk in the private key matrix of IPK key generation protocol; Second, the influence of matrix size in IPK key generation protocol on security is analyzed; Third, in the IPK key generation protocol the security of the declared public key is checked, that is, whether the public key has the risk of replacement attack; Finally, we need to ensure that the composite private key is collision resistant.

### 3.1 Security of private key

**3.1.1 Security of private key matrix.** **Theorem 3.1**. *If the one-time encryption OTP (one-time password) is secure, then the security of the private key matrix in the IPK algorithm is ensured.*

The first-order composite private key $isk'$ obtained by the attacker from the key center consists of two parts: the identity private key $isk_m$ and the private key $r_{i,2}$ randomly generated by the key center, namely $isk' = isk_m + r_{i,2}$, where the key center randomly generates new $r_{i,2}$ every time, and they are different.

Suppose that the private key matrix is $s_k = (a_{ij})_{m \times h}$. According to the private key generation algorithm, the key center takes the attacker ID $ID_i$ as the data and the declared public key $R_{A_i}$ as the private key to calculate the HMAC value based on SM3, and maps the value to the private key matrix, and then he sum of the obtained $h$ numbers is recorded as the matrix ID private key, i.e.,

$$msk_i = x_1 + x_2 + \cdots + x_h.$$

The final output private key of the key center is

$$isk' = msk_i + r_{i,2} = x_1 + x_2 + \cdots + x_h + r_{i,2}.$$

Let

$$x_k = \begin{pmatrix} t_1 & \cdots & t_m \end{pmatrix}_k \begin{pmatrix} a_{k,1} \\ \vdots \\ a_{k,m} \end{pmatrix},$$

where $t_1, t_2, \cdots, t_m \in \{0, 1\}$. We thus rewritten $msk_i$ by

$$
msk_i = (\, t_1 \quad \cdots \quad t_m \,)_1 \begin{pmatrix} a_{1,1} \\ \vdots \\ a_{1,m} \end{pmatrix} + (\, t_1 \quad \cdots \quad t_m \,)_2 \begin{pmatrix} a_{2,1} \\ \vdots \\ a_{2,m} \end{pmatrix}
$$

$$
+ \cdots + (\, t_1 \quad \cdots \quad t_m \,)_h \begin{pmatrix} a_{m,1} \\ \vdots \\ a_{m,h} \end{pmatrix}.
$$

By the principle of private key generation, a total of $m^h$ distinct private keys can be generated from the private key matrix. Subsequently, the equations based on all the private keys are listed as follows:

$$
\begin{pmatrix} t_{1(1,1)} & \cdots & t_{1(m,1)} & \cdots & t_{1(m,h)} \\ t_{2(1,1)} & \cdots & t_{2(m,1)} & \cdots & t_{2(m,h)} \\ \vdots & \vdots & \vdots & \vdots & \vdots \\ t_{m^h(1,1)} & \cdots & t_{m^h(m,1)} & \cdots & t_{m^h(m,h)} \end{pmatrix} \cdot \begin{pmatrix} a_{1,1} \\ a_{1,2} \\ \vdots \\ a_{1,h} \\ \vdots \\ a_{m,1} \\ a_{m,2} \\ \vdots \\ a_{m,h} \end{pmatrix} = \begin{pmatrix} msk_1 \\ msk_2 \\ \vdots \\ msk_h \\ \vdots \\ msk_{m^{h-1}+1} \\ msk_{m^{h-1}+2} \\ \vdots \\ msk_{m^h} \end{pmatrix}.
$$

We can simply denote the above equation by $T \cdot SK = MSK$, where

$$
T = \begin{pmatrix} t_{1(1,1)} & \cdots & t_{1(m,1)} & \cdots & t_{1(m,h)} \\ t_{2(1,1)} & \cdots & t_{2(m,1)} & \cdots & t_{2(m,h)} \\ \vdots & \vdots & \vdots & \vdots & \vdots \\ t_{m^h(1,1)} & \cdots & t_{m^h(m,1)} & \cdots & t_{m^h(m,h)} \end{pmatrix} \in \mathbb{Z}_q^{m^h \times mh},
$$

$$
SK = \begin{pmatrix} a_{1,1} \\ a_{1,2} \\ \vdots \\ a_{1,h} \\ \vdots \\ a_{m,1} \\ a_{m,2} \\ \vdots \\ a_{m,h} \end{pmatrix} \in \mathbb{Z}_q^{mh}, MSK = \begin{pmatrix} msk_1 \\ msk_2 \\ \vdots \\ msk_h \\ \vdots \\ msk_{m^{h-1}+1} \\ msk_{m^{h-1}+2} \\ \vdots \\ msk_{m^h} \end{pmatrix} \in \mathbb{Z}_q^{m^h}.
$$

Clearly, there are $m^h$ different private keys $msk_i$ generated by the private key matrix. All possible private keys $msk_i$ consist the matrix $MSK$.

The objective of the attacker is to solve $SK$ under the relation $T \cdot SK = MSK$. Nan [22] has proved that the general solution $\widetilde{SK}$ of the private key matrix can be found from the relation, which has the same effect as the original private key matrix. To counteract the attack, this method incorporates the randomly generated key $r_{i,2}$ from the key center. This prevents attackers from deducing the correspondence mentioned above and restricts their access to only obtaining the key $isk' = msk_i + r_{i,2}$, where $i = 1, 2, \ldots, m^h$.

We simply let $T \cdot SK + R = ISK$, where

$$
R = \begin{bmatrix} r_{1,2} \\ r_{2,2} \\ \vdots \\ r_{h,2} \\ \vdots \\ r_{m^{h-1}+1,2} \\ r_{m^{h-1}+2,2} \\ \vdots \\ r_{m^h,2} \end{bmatrix} \in Z_q^{m^h}, \quad ISK = \begin{bmatrix} isk'_1 \\ isk'_2 \\ \vdots \\ isk'_h \\ \vdots \\ isk'_{m^{h-1}+1} \\ isk'_{m^{h-1}+2} \\ \vdots \\ isk'_{m^h} \end{bmatrix} \in Z_q^{m^h}.
$$

This problem can be transformed into adding noise $R$ to the original matrix relationship, which destroys the original linear relationship $T \cdot SK = MSK$, and the Gaussian elimination method is invalid, so there is no effective algorithm to solve it.

Since $R$ is random sampling, then entropy $H(SK|ISK) = H(SK)$, which means that even knowing $ISK$ does not increase the attacker's knowledge of the private key matrix $SK$, so our method has perfect confidentiality. The system can be regarded as a one-time encryption OTP, where $R$ is the random key of the OTP. If an attacker obtains $SK$ (or $T \cdot SK$) from $ISK$ without knowing $R$, the attacker can crack any OTP system, which contradicts the known reality, so the private key matrix in the IPK is secure and resistant to linear collusion attacks.

**3.1.2 Security of the second-order composite private key.** The hardness of the elliptic curve discrete logarithm problem (ECDLP) [30, 31] is crucial for the security of elliptic curve cryptographic schemes. Under some well known conditions, such as the ECDLP is intractable against all known attacks [32].

**Theorem 3.2**. *If the ECDLP problem is hard, the second-order composite private key of the IPK algorithm is secure.*

Suppose there exists an attacker $A_1$ who impersonates an honest and curious key center and wants to try to obtain the second-order combined private key (equivalent to obtaining the private key of the user ID). We can formally define the behavior of the attacker, as following:

1. Alice randomly generates a key pair $(r_1, R_1)$ that satisfies $R_1 = [r_1]G$;

2. Alice sends $R_1$ and the ID to attacker $A_1$;

3. Attacker $A_1$ randomly choose $(r_2, R_2)$, where $R_2 = [r_2]G$, use the ID of Alice and the declared public key $R = R_1 + R_2$ to generate the declared private key $isk_m$, and then get the first-order composite private key $isk' = (isk_m + r_2) \bmod n$. In the view of attacker, the user's public key

$R_1$, the first-order composite private key $isk'$ and the second-order composite signature public key $IPK$ can be obtained.

The final signature private key of Alice is the second-order compound private key $isk = isk' + r_1$, and the corresponding public key is $IPK$. There are two ways for attacker $A_1$ to obtain the user's private key $isk$, from $R_1$ or $IPK$ to resolve the private key, which is difficult since the ECDLP problem is hard.

## 3.2 Private key matrix size and its security

The quantitative security of a cryptographic scheme is called the security strength or security level (hereinafter referred to as the security strength). It usually refers to the computation cost of solving a hard problem when the cryptographic scheme takes a certain parameter under the current best effective attack method.

The following definition is security strength of public key cryptographic schemes.

**Definition 1**. *Suppose the time required to determine the key of a public key cryptosystem is similar to the time needed to determine the key of a symmetric cryptosystem. In this case, the security strength of the public key cryptosystem is equal to the key length of the symmetric cryptosystem.*

The security of the private key matrix is based on the relies on the numerous possibilities of partitions of large integer. In integer partitions [33], the following lemmas are obvious.

**Lemma 3.1**. *Suppose that $1 \leq v_i \leq \ell$ for each $i \in \{1, 2, \ldots, t\}$ and $\ell \in \mathbb{Z}$. The following equation*

$$v_1 + v_2 + \cdots + v_t = s$$

*with $s = \left[\frac{1+\ell}{2} t\right]$ has the largest number of solutions.*

**Lemma 3.2**. *Let $v_1$, $v_2$ be integers on interval $[1, \ell]$, where $\ell \in \mathbb{Z}$. We have*

$$P(v_1 + v_2 = 1 + \ell) = \frac{2}{\ell - 1}.$$

*When $\ell$ is large enough, e can approximate the probability as $2/\ell$. Moreover, if there are $\ell'$ integers in $[1, \ell]$ which can not be selected, we have*

$$P(v_1 + v_2 = 1 + \ell) \leq \frac{2}{\ell - \ell' - 1}.$$

**Lemma 3.3**. *If all integers $v_i \in [1, \ell]$ are randomly chosen, then we have*

$$P\left(v_1 + v_2 + \cdots + v_t = \left[\frac{1+\ell}{2} t\right]\right) \leq P\left(v_1 + v_2 + \cdots + v_{t-1} = \left[\frac{1+\ell}{2} (t-1)\right]\right).$$

*The maximum probability is given at $t = 2$, which is Lemma 2.*

**Theorem 3.3**. *For $m$, $h \geq 32$, the private key matrix has collision resistance.*

*Proof.* Let $u_1, u_2, \ldots, u_h$ be the array formed by extracting one element per column from the $mh$ matrix the first time and $v_1, v_2, \ldots, v_h$ the second time. After a matrix $M$ is chosen, the probability of collision under this matrix can be denoted as

$$P(v_1 + v_2 + \cdots + v_h = u_1 + u_2 + \cdots + u_h, u_i, v_i \text{ are not all the same}|M).$$

We denote event $B = \{v_1 + v_2 + \cdots + v_h = u_1 + u_2 + \cdots + u_h, u_i, v_i \text{ are not all the same}\}$. By the

total probability formula, the average collision probability is

$$P(B) = \sum_M P(M)P(B|M).$$

It should be noted that $B$ in this case does not mean that $u_i$, $v_i$ are chosen randomly. The process should be to randomly select $u_1, u_2, \ldots, u_h$. Next, we give $u_1, \ldots, u_h$ with $m-1$ numbers each and let them form the $h$ columns of the matrix, which we denote as $A(u_1, u_2, \ldots, u_h)$, abbreviated as $A_u$. Obviously, apart from the identified elements $u_1, \ldots, u_h$ in $A_u$, the remaining numbers are filled in $A_u$ randomly. At this point, $v_1, v_2, \ldots, v_h$ are extracted from $A_u$. Note that the probability that $v_i$ chooses $u_i$ is $1/m$; The probability that $v_i$ choose the other possible number is equally divided $(m-1)/m$. Hence,

$$
\begin{aligned}
P(B) \quad &= \sum_s P(u_1 + u_2 + \cdots + u_h = s) \\
&\quad \times P(v_1 + v_2 + \cdots + v_h = s, u_i, v_i \text{ are not all the same} | u_1 + u_2 + \cdots + u_h = s) \\
&= \sum_{\substack{s \\ u_1, u_2, \ldots, u_h}} P(u_1 + u_2 + \cdots + u_h = s) \\
&\quad \times \sum_{A_u} P(A_u) P(v_1 + v_2 + \cdots + v_h = s, u_i, v_i \text{ are not all the same} | A_u).
\end{aligned}
$$

We now focus on

$$\mathcal{P} = \sum_{A_u} P(A_u) P(v_1 + v_2 + \cdots + v_h = s, u_i, v_i \text{ are not all the same} | A_u).$$

According to the requirements of selection, we have

$$
\begin{aligned}
\mathcal{P} \quad &= \sum_{t \neq u_h} P(v_h = t) P(v_1 + \cdots + v_{h-1} = s - t) \\
&\quad + \frac{1}{m} P(v_1 + \cdots + v_{h-1} = s - u_h, u_i, v_i \text{ are not all the same}).
\end{aligned}
$$

Let

$$\mathcal{L}_{h-1} = \sum_{t \neq u_h} P(v_h = t) P(v_1 + \cdots + v_{h-1} = s - t);$$

$$\mathcal{H}_{h-1} = P(v_1 + \cdots + v_{h-1} = s - u_h, u_i, v_i \text{ are not all the same}).$$

Since $t \neq u_h$ in $\mathcal{L}_{h-1}$ already satisfies the condition that $u_i$, $v_i$ are not all identical, this condition is no longer mandatory. For two groups to sum equally, at most $h-2$ of the remaining $h-1$ numbers $v_i$ are the same with the corresponding $u_i$. So for the sake of notation, we write the same numbers backwards, starting with $u_{h-1}$. It follows that

$$
\begin{aligned}
\mathcal{L}_{h-1} \quad &= \sum_{t \neq u_h} P(v_h = t) \left( \binom{h-1}{h-2} \left(\frac{1}{m}\right)^{h-2} P(v_1 = s - t - u_{h-1} - \cdots - u_2) \right. \\
&\quad \left. + \sum_{i=2}^{h-1} \binom{h-1}{h-i-1} \left(\frac{1}{m}\right)^{h-i-1} P(v_1 + \cdots + v_i = s - t - u_{h-1} - \cdots - u_{i+1}) \right).
\end{aligned}
$$

Note that $v_1 = s - t - u_{h-1} - \cdots - u_2$ is a determined constant. Hence,

$$
\begin{aligned}
&\sum_{t \neq u_h} P(v_h = t) \binom{h-1}{h-2} \left(\frac{1}{m}\right)^{h-2} P(v_1 = s - t - u_{h-1} - \cdots - u_2) \\
&< \sum_{t \neq u_h} P(v_h = t) \binom{h-1}{h-2} \left(\frac{1}{m}\right)^{h-2} \frac{1}{m} = \left(\frac{1}{m}\right)^{h-2} \frac{(m-1)(h-1)}{m^2}.
\end{aligned}
\tag{1}
$$

For probability $P_i = P(v_1 + \cdots + v_i = s - t - u_{h-1} - \cdots - u_{i+1})$, random selections of $v_1, v_2, \ldots,$ $v_i$ can be made from at least $2^{256} - mh$ numbers. By Lemmas 1, 2 and 3, we derive

$$
\begin{aligned}
P_i \quad &\leq P\left(v_1 + \cdots + v_i = \left[\frac{2^{256} + 1}{2} i\right]\right) \\
&\leq P(v_1 + v_2 = 1 + 2^{256}) \\
&\leq \frac{2^{257}}{\left(2^{256} - mh\right)^2} \approx \frac{1}{2^{255}}.
\end{aligned}
$$

In addition, we usually take $m \geq h \geq 32$. Then we can get

$$
\sum_{i=2}^{h-1} \binom{h-1}{h-i-1} \left(\frac{1}{m}\right)^{h-i-1} < \sum_{i=2}^{h-1} \frac{1}{i!} < 1.
$$

Thus,

$$
\sum_{t \neq u_h} P(v_h = t) \sum_{i=2}^{h-1} \binom{h-1}{h-i-1} \left(\frac{1}{m}\right)^{h-i-1} P_i < \frac{m-1}{m 2^{255}}.
\tag{2}
$$

Combining Eqs (1) and (2) together we obtain

$$
\mathcal{L}_{h-1} < \left(\frac{1}{m}\right)^{h-2} \frac{(m-1)(h-1)}{m^2} + \frac{m-1}{m 2^{255}} < \max\left\{h\left(\frac{1}{m}\right)^{h-1}, \frac{1}{2^{255}}\right\}.
$$

We now try to give a reasonable upper bound for $\mathcal{H}_{h-1}$. Let $s' = s - u_h$. We get

$$
\begin{aligned}
\mathcal{H}_{h-1} \quad &= \sum_{t \neq u_{h-1}} P(v_{h-1} = t) P(v_1 + \cdots + v_{h-2} = s' - t) \\
&\quad + \frac{1}{m} P(v_1 + \cdots + v_{h-2} = s' - u_{h-1}) \\
&= \mathcal{L}_{h-2} + \frac{1}{m} \mathcal{H}_{h-2}.
\end{aligned}
$$

Iterating the above equation repeatedly we deduce that

$$
\begin{aligned}
\mathcal{H}_{h-1} \quad &< \left(\frac{1}{m}\right)^{h-2} (h-1) + \frac{1}{2^{255}} + \frac{1}{m} \mathcal{H}_{h-2} \\
&< \cdots \\
&< \left(\frac{1}{m}\right)^{h-2} (2 + 3 + \cdots + h - 1) + \frac{1}{2^{254}} + \left(\frac{1}{m}\right)^{h-3} \mathcal{H}_2.
\end{aligned}
$$

Clearly, $\mathcal{H}_2 = \sum_{v_1 \neq u_1, v_2 \neq u_2} P(v_1 + v_2 = s - u_3 - \cdots - u_h) \leq P(v_1 + v_2 = 2^{256} + 1) < \frac{1}{2^{254}}$.

**Table 2. Proposed parameters for private key matrix.**

| $m$ | $h$ | Security strength | Equivalent symmetric cipher | Private key |
|-----|-----|-------------------|------------------------------|-------------|
| 32  | 32  | 160B              |                              |             |
| 64  | 32  | 192B              | AES                          | 192B        |
| 128 | 32  | 224B              |                              |             |
| 256 | 32  | 256B              | AES                          | 256B        |

Recall that $m, h \geq 32$. We directly get

$$\mathcal{H}_{h-1} \quad < \left(\frac{1}{m}\right)^{h-2} (2 + 3 + \cdots + h - 1) + \left(\left(\frac{1}{m}\right)^{h-3} + 1\right) \frac{1}{2^{254}}$$

$$< \left(\frac{1}{m}\right)^{h-2} \frac{h^2}{2} + \frac{1}{2^{254}}.$$

We then have

$$\mathcal{P} \quad = \mathcal{L}_{h-1} + \frac{1}{m}\mathcal{H}_{h-1}$$

$$< \left(\frac{1}{m}\right)^{h-1} \left(\frac{h^2}{2} + h\right) + \frac{1}{2^{254}}.$$

We finally derive

$$P(B) = \sum_{\substack{s \\ u_1, u_2, \ldots, u_h}} P(u_1 + u_2 + \cdots + u_h = s)\mathcal{P}$$

$$< \left(\frac{1}{m}\right)^{h-1} \left(\frac{h^2}{2} + h\right) + \frac{1}{2^{254}}.$$

As both $m$ and $h$ increase, $P(B)$ diminishes progressively. Therefore, the maximum value of the upper bound of $P(B)$ realizes at $m = h = 32$, i.e.,

$$P(B) < (2^5 + 2^9)\frac{1}{2^{155}} + \frac{1}{2^{254}} < \frac{1}{2^{145}}.$$

This probability is small enough so that our private key matrix is collision resistant.

As shown above, the security strength depends on the choice of $m$ and $h$. This paper provides four sets of proposed parameters, which are (1) $m = 32$, $n = 32$; (2) $m = 64$, $n = 32$; (3) $m = 128$, $n = 32$; (4) $m = 256$, $n = 32$. Their security strength is shown in Table 2.

## 3.3 Non-existence of substitution attacks

Suppose that the attacker A has the ability to replace any user ID and public key $R$ with any value, and the goal of attack is to generate the same composite public key IPK as the original user. Attacker A and challenger C engage in the following game:

1. System initialization: The challenger C runs the system to obtain the public key matrix PKM and sends the PKM to attacker A;

2. Claim public key query: Attacker A can query any claim public key with identity $ID_i$. In response, the challenger returns the claim public key of $ID_i$;

3. Output: Attacker A outputs $(ID_B, R_B, IPK_A)$, where the declared public key of $ID_A$ has been queried.

If the challenger C checks that the output of the attacker A satisfies the composite public key generated by $(ID_B, R_B)$ is $IPK_A$, it means that the attacker A challenges successfully.

**Theorem 3.4**. *In the random oracle model, if the SM3 algorithm has one-way and collision resistance, IPK is resistant to substitution attack.*

*Proof*. If an attacker A can use $(ID_B, R_B)$ to produce a composite public key $IPK_A$, the identifying public key is $IPK_m = IPk_A - R_B$. According to the calculation method of the public key, the challenger C takes the identity $ID_B$ as the data and declares the public key $R_B$ as the key to calculate the HMAC based on SM3 and obtain the hash value $h$ of 32 bytes. Noting that the hash value uniquely determines the identifying public key, let us assume that the hash value corresponding to the identifying public key $IPK_m$ is $h_1$, thereby obtaining the relation:

$$SM3_{R_B}(ID_B) = h_1.$$

If the attacker can generate $(ID_B, R_B)$ that meets the requirements, the plaintext and key corresponding to any SM3 hash value can be obtained, which is contrary to the conclusion that SM3 hash algorithm has one-way and collision resistance, so the IPK scheme in this paper is resistant to substitution attack.

## 3.4 Security of composite private keys

In the previous subsections, we considered the security of each part of the composite private key. This subsection is mainly to consider the collision problem of composite private keys.

**Theorem 3.5**. *Given a random integer in the interval $[0, 2^N - 1)$, the probability of guessing correctly is $1/2^N$, which implies that the identification cost is $O(2^N)$. In other words, if the probability of collision depends on the length of the number, then it is an NP problem.*

Next, we illustrate our scheme to generate composite private key is an NP problem. Recall that the order of a base point in the SM2 algorithm is never lower than $\alpha 2^N$, where $\alpha > 0$.

**Theorem 3.6**. *The collision probability of the composite private key does not exceed $1/\alpha 2^N$. Therefore, the cracking cost for IPK is at least $O(2^N)$, so breaking the composite private key is NP-hard.*

*Proof*. Consider the first extracted composite private key $isk = u_1 + u_2 + \cdots + u_h + r_1 + r_2$, where $isk$ is used as a variable to iterate over all possible values. If the re-extracted composite private key $isk^* = v_1 + v_2 + \cdots + v_h + \gamma_1 + \gamma_2$ collids with $isk$, the collision probability is $\mathcal{S} = \sum_{isk} P(isk^* = isk)$. Note that the two extraction is random, have independence. By total probability formula, we have

$$\mathcal{S} = \sum_{isk} P(u_1 + u_2 + \cdots + u_h + r_1 + r_2 = isk) P(v_1 + v_2 + \cdots + v_h + \gamma_1 + \gamma_2 = isk).$$

We focus on the value of $\mathcal{S}' = P(v_1 + v_2 + \cdots + v_h + \gamma_1 + \gamma_2 = isk)$. Let $s = \gamma_1 + \gamma_2$. Using total probability formula again, we obtain

$$\mathcal{S}' = \sum_{\gamma_1 + \gamma_2 = s} P(v_1 + v_2 + \cdots + v_h = isk - s | \gamma_1 + \gamma_2 = s) P(\gamma_1 + \gamma_2 = s).$$

Note that the sum of $v_1, \ldots, v_h$ must equal $isk - s$ if $\gamma_1, \gamma_2$ are chosen well. But, the choice of $v_1, \ldots, v_h$ depends on the declared public key, which is the hash value obtained by SM3-based

HMAC and thus can be regarded as independent of $\gamma_1, \gamma_2$. It follows that

$$P(v_1 + v_2 + \cdots + v_h = isk - s | \gamma_1 + \gamma_2 = s) = P(v_1 + v_2 + \cdots + v_h = isk - s).$$

Hence,

$$\mathcal{S}' = \sum_{\gamma_1 + \gamma_2 = s} P(v_1 + v_2 + \cdots + v_h = isk - s) P(\gamma_1 + \gamma_2 = s).$$

Recall that the order $n$ of the base point is no less than $\alpha 2^N$. We we have selected $\gamma_1$, the only choice for $\gamma_2$ is $s - \gamma_1$, which implies that $P(\gamma_1 + \gamma_2 = s) \leq 1/\alpha 2^N$. We then have

$$\mathcal{S}' \leq \sum_{\gamma_1 + \gamma_2 = s} P(v_1 + v_2 + \cdots + v_h = isk - s) \frac{1}{\alpha 2^N} \leq \frac{1}{\alpha 2^N}.$$

Therefore,

$$\mathcal{S} \leq \frac{1}{\alpha 2^N} \sum_{isk} P(u_1 + u_2 + \cdots + u_h + r_1 + r_2 = isk) = \frac{1}{\alpha 2^N}.$$

In summary, we find that the probability of collision is not more than $1/\alpha 2^N$, as desired.

**Remark 1**. *In the SM2 algorithm, $N = 256$ is typically employed. Consequently, the collision probability of our composite private key will not exceed $1/\alpha 2^{256}$, and as $\alpha$ is not overly small, this collision probability is generally less than $1/2^{200}$.*

Generally speaking, such a small-probability collision problem can be transformed into a circuit satisfiability problem [34], which belongs to the NP category. Therefore, the composite private key cracking problem is also an NP problem.

## 4 Technical comparison

Several management methods in the certificateless public key management system based on identification, such as TF-CPK, IBC(SM9), manage public keys through identification, which simplifies the management complexity of public keys and has many technical similarities. However, due to different methods in the implementation of public key management system and key application, there are differences in technical indicators. In the following, IPK is compared and analyzed with several major identity cryptosystems, please refer to Table 3.

TF-CPK, IPK and SM2 implicit certificates use the same SM2 cryptographic algorithm, while the cryptographic algorithm used by IBC is different from SM2. In order to have comparability, the analysis and comparison are based on the security strength of $2^{128}$ as the security

**Table 3. IPK compared with IBC(SM9), TF-CPK, and SM2 implicit certificates.**

| Identity-based cryptosystem | IBC(SM9) | TF-CPK | SM2 implicit certificates | IPK |
|---|---|---|---|---|
| Length of signed private key | 64B | 32B | 32B | 32B |
| Length of signed public key | Arbitrary | 64B | 64B | 64B |
| Length of private key encryption | 128B | 32B | 32B | 32B |
| Length of public key encryption | Arbitrary | 64B | 64B | 64B |
| Key encapsulation | 96B | 128B | 128B | 128B |
| Signature length | 96B | 128B | 128B | 128B |
| Signature time | 12 + 1UT | 1UT | 1UT | 1UT |
| Verification time | 12UT | 2 + 2UT | 1 + 2UT | 2UT |
| Storage space | 128 + 64B | 64KB | 64B | 64KB |

baseline. In the performance comparison, the unit time (denoted as UT) is taken as the operation time of one 256-bit elliptic curve multiple point. We now briefly explain how the data in Table 3 came from.

1. We initially indicate the length of the digital signature.

a. The signature of SM9 has the form ($h$, $S$) and the length of $h$ and $S$ are 32B and 64B, respectively. So the total length of signature of SM9 is 96B;

b. TF-CPK consists of the signature data ($r$, $s$), the declared public key, and the signature of the identity to the declared public key. Thus, its length is 64B × 3 = 192B;

c. The signature of SM2 implicit certificates is composed by the signature data ($r$, $s$) and the declared public key, and the signature length is 128B in total;

d. In terms of digital signatures, IPK has the identical structure to SM2 implicit certificates, and thus its signature length is 128B.

2. Assuming that the private key has a length 32B, the key encapsulation length is as follows.

a. The output of the private key encapsulation of SM9 takes the form ($K$, $C$), where the length of $K$ and $C$ are 32B and 64B, respectively. So the total length is 96B;

b. TF-CPK, SM2 implicit certificate, and IPK all employ the SM2 cryptosystem, thus their length is 128B.

3. Regarding the computational cost, the main time consumption of the algorithm performance encompassed in several schemes is concentrated on the multi-point operation of the elliptic curve. Hence, the time cost of calculation is compared and analyzed with one multi-point operation as the benchmark time unit UT.

a. SM9 digital signature requires 13UT (one operation of addition group and one operation of multiplication group), and verification signature is 12UT (one operation of multiplication group);

b. TF-CPK is based on SM2. The time cost of digital signature takes 1UT (one operation of addition group), and the verification signature involves data signature verification and public key signature verification, which needs 4UT (2 times of signature verification, 2 operations of addition group for each signature verification);

c. The SM2 implicit certificate is also based on SM2. The time cost of digital signature is 1UT (one operation of addition group), the verification signature includes public key calculation and data signature verification, where the calculation of public key has one multi-point operation and the data signature verification needs 2UT;

d. IPK is also based on SM2. Similarly, the time cost of digital signature is 1UT. The verification signature includes public key calculation and data signature verification. However, computing the public key merely involves simple operations, and the time consumption is negligible, thus it takes 2UT to verify the signature.

As shown in Table 3, the computation cost of SM9 cryptographic algorithm is significantly higher than that of several identity-based public key management schemes based on SM2 cryptographic algorithm, and IPK has the least computation cost. However, the storage cost of IPK and TF-CPK is larger than that of other schemes, which is not suitable for embedded terminal devices with very limited storage space.

## 5 Conclusion

In this paper, we introduce the IPK scheme and verify its security. In summary, the innovations of IPK key generation protocol mainly include the following contents:

1. IPK uses the public and private key matrix and mapping method to realize the binding of identity and key, which can simplify the distribution management of public key through identity;

2. In the process of private key generation, both the center and the terminal participate in the final composite generation of the terminal, that is, a random factor is added in the process of key production to eliminate the linear correlation between the private keys;

3. The terminal private key is finally composite generated by the terminal, and the center does not know the terminal's second-order composite private key, which solves the trust problem that the identification key can only be generated by the center, and the terminal private key cannot be obtained by the key center in this cooperative generation method, so that the signature can be in line with the electronic signature method;

4. The declared public key factor $R$ directly participates in the identity mapping algorithm to realize the binding relationship between the declaring public key and the identity, and there is no risk of replacement attack.

IPK generates a large number of keys through a smaller matrix and the idea of combinatorial mathematics. It is a lightweight identification key generation protocol, which simplifies the complexity of key generation and reduces the construction cost and operation cost of key system.

Theoretically, the rigorous determination of the collision probability can assist us in reducing the size of the random matrix. As part of our future work, we will try to decrease the upper bound on the collision probability using mathematical tools. Moreover, we will elaborate on the possible applications of IPK, such as cloud computing, e-health and Internet of things.

## Acknowledgments

We thank the anonymous reviewers for their valuable feedback.

## Author Contributions

**Conceptualization:** Cheng Zeng.

**Formal analysis:** Xin Sun, Cheng Zeng.

**Funding acquisition:** Xin Sun.

**Methodology:** Bang Lv, Changhua Sun.

**Validation:** Jiajia Han.

**Visualization:** Changhua Sun.

**Writing – original draft:** Cheng Zeng.

**Writing – review & editing:** Jiajia Han, Bang Lv, Changhua Sun.

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
