## [Decision Letter · Decision Letter 0]

17 Sep 2024

PONE-D-24-38257A novel method for generating public keys involving matrix operationsPLOS ONE

Dear Dr. Zeng,

Thank you for submitting your manuscript to PLOS ONE. After careful consideration, we feel that it has merit but does not fully meet PLOS ONE’s publication criteria as it currently stands. Therefore, we invite you to submit a revised version of the manuscript that addresses the points raised during the review process.

We look forward to receiving your revised manuscript.

Kind regards,

Zhiquan Liu, Ph.D.

Academic Editor

PLOS ONE

Journal Requirements:

Please confirm at this time whether or not your submission contains all raw data required to replicate the results of your study. Authors must share the “minimal data set” for their submission. PLOS defines the minimal data set to consist of the data required to replicate all study findings reported in the article, as well as related metadata and methods (https://journals.plos.org/plosone/s/data-availability#loc-minimal-data-set-definition). For example, authors should submit the following data: - The values behind the means, standard deviations and other measures reported; - The values used to build graphs; - The points extracted from images for analysis. Authors do not need to submit their entire data set if only a portion of the data was used in the reported study. If your submission does not contain these data, please either upload them as Supporting Information files or deposit them to a stable, public repository and provide us with the relevant URLs, DOIs, or accession numbers. For a list of recommended repositories, please see https://journals.plos.org/plosone/s/recommended-repositories. If there are ethical or legal restrictions on sharing a de-identified data set, please explain them in detail (e.g., data contain potentially sensitive information, data are owned by a third-party organization, etc.) and who has imposed them (e.g., an ethics committee). Please also provide contact information for a data access committee, ethics committee, or other institutional body to which data requests may be sent. If data are owned by a third party, please indicate how others may request data access.

4. We note you have included a table to which you do not refer in the text of your manuscript. Please ensure that you refer to Table 1 and 2 in your text; if accepted, production will need this reference to link the reader to the Table.

Additional Editor Comments:

All the 3 reviewers hold a positive attitude towards this paper and point out some existing problems in this paper. The author is requested to make careful revisions according to the review comments, and then submit its revised version.

Reviewers' comments:

Reviewer's Responses to Questions

**Comments to the Author**

1. Is the manuscript technically sound, and do the data support the conclusions?

Reviewer #1: Yes

Reviewer #2: Yes

Reviewer #3: Yes

2. Has the statistical analysis been performed appropriately and rigorously? 

Reviewer #1: Yes

Reviewer #2: Yes

Reviewer #3: Yes

3. Have the authors made all data underlying the findings in their manuscript fully available?

Reviewer #1: Yes

Reviewer #2: Yes

Reviewer #3: Yes

4. Is the manuscript presented in an intelligible fashion and written in standard English?

Reviewer #1: Yes

Reviewer #2: Yes

Reviewer #3: Yes

5. Review Comments to the Author

Reviewer #1: 1. Good research work

2. The proposed work explained very well and ordered correctly

3. All the results and discussions are very well.

4. The work explained very well.

5. Check the reference are cited correctly.

Reviewer #2: The paper is well-written and presents a new method to generate an identity key (IPK). It is based on SM2 elliptic curve cryptography and random matrix theory. The paper is technically sound and can be accepted as it is for publication.

Reviewer #3: In this paper, the authors propose a new approach to generate an identity key, named identity public key (IPK). IPK identity key generation protocol is based on SM2 elliptic curve cryptography and random matrix theory. It builds upon the concept of combined public key (CPK) and resolves the linear collusion issue of CPK as well as the authenticity verification problem of the declared public key of the simplified TF-CPK by enhancing the identity mapping approach. I think it's good, but there are a few problems.

(1) The abstract of the paper is too brief, and it is suggested to add some content. (2) Table headings are recommended for the top of the table, not the bottom. (3) The discussion about recent related schemes, such as efficient privacy-preserving spatial range query over outsourced encrypted data, time-controllable keyword search scheme with efficient revocation in mobile e-health cloud, efficient privacy-preserving spatial data query in cloud computing, instead of conventional schemes are suggested. (4) In the conclusion part, it is suggested to add the expression of future work. (5) It is suggested that the author increase the function and performance comparison with other schemes. (6) References should not appear in the abstract.

6. PLOS authors have the option to publish the peer review history of their article (what does this mean?). If published, this will include your full peer review and any attached files.

Reviewer #1: No

Reviewer #2: **Yes: **Gaurav Mittal

Reviewer #3: No

---

## [Author Response · Author response to Decision Letter 0]

10 Oct 2024

Thank you for your valuable suggestions and comments. Please see the PDF named 'Response to reviewers' for more details. We hope our changes can meet the reviewer’s requirements and lead to the publication of the manuscript.

---

## [Editor Report · Decision Letter 1]

11 Oct 2024

A novel method for generating public keys involving matrix operations

PONE-D-24-38257R1

Dear Dr. Zeng,

We’re pleased to inform you that your manuscript has been judged scientifically suitable for publication and will be formally accepted for publication once it meets all outstanding technical requirements.

Kind regards,

Zhiquan Liu, Ph.D.

Academic Editor

PLOS ONE

Additional Editor Comments (optional):

Accept
---

## [Editor Report · Acceptance letter]

17 Oct 2024

PONE-D-24-38257R1 

PLOS ONE

Dear Dr. Zeng, 

I'm pleased to inform you that your manuscript has been deemed suitable for publication in PLOS ONE. Congratulations! Your manuscript is now being handed over to our production team.

Kind regards, 

on behalf of

Professor Zhiquan Liu 

Academic Editor

PLOS ONE